# Inverse Gaussian distribution of wave set-up heights along a shoreline with complicated geometry

Tarmo Soomere<sup>1,2</sup>, Katri Pindsoo<sup>1</sup>

<sup>1</sup>Laboratory of Wave Engineering, Institute of Cybernetics at Tallinn University of Technology, Akadeemia tee 21, Tallinn, 12618, Estonia

<sup>2</sup>Estonian Academy of Sciences, Kohtu 6, Tallinn, 10130, Estonia

*Correspondence to*: Katri Pindsoo (katri.pindsoo@ioc.ee)

Abstract. The phenomenon of wave set-up may substantially contribute to the formation of devastating coastal flooding in certain coastal sections. We study empirical probability distribution of the occurrence of different set-up heights in section of

- coastline near Tallinn in the Gulf of Finland, the eastern Baltic Sea. The shoreline in the study area is often attacked by high waves from various directions and also has a complex geometry. Shown is that this distribution substantially deviates from the Rayleigh or the Weibull distribution that usually reflect the distribution of different wave heights. The distribution of wave set-up heights matches a Wald (inverse Gaussian) distribution along the entire study area. Even though different sections of the study area are open to different directions and host substantially different wave regimes, the leading term of
- the exponent in the associated inverse Gaussian distribution varies insignificantly along the study area and generally is close to -1. This appearance signals that extreme set-up events are substantially more probable that it could be expected from the probability of occurrence of severe seas. This feature is invariant with respect to the orientation of the coastline and with respect to the properties of local wave climate.

# **1** Introduction

- The increase in the global sea level in current and projected marine climate change (Cazenave et al., 2014) is often associated with major consequences (Hallegatte et al., 2013) and economic damages to low-lying coastal areas (Darwin and Tol, 2001) that may lead to a loss of worldwide welfare of almost by 2% by the end of this century (Pycroft et al., 2016). This increase, however, contributes only a small fraction into the most devastating coastal floodings. These events, additionally to being economically extremely damaging (Meyer et al., 2013), may also lead to massive losses of lives and desertification of entire coastal communities (Dube et al., 2009).
- A devastating flooding is usually caused by the interplay of several drivers with fundamentally different predictability, physical, dynamical and statistical properties, and with different level of correlations between their contributions. For example, tides are almost perfectly regular and caused by extra-terrestrial drivers while a reasonable forecast of the impact of low atmospheric pressure (inverted barometric effect), wind-driven surge and wave-induced set-up requires a cluster of
- dedicated atmospheric, ocean circulation and wave models. The resulting high water levels may be additionally amplified by

specific events and mechanisms such as tide-surge interactions (Batstone et al., 2013; Olbert et al., 2013), meteorologically driven long waves (Pattiarachi and Wijeratne, 2014; Pellikka et al., 2014; Vilibic et al., 2014) or seiches (Vilibic, 2006; Kulikov and Medvedev, 2013).

Along with contemporary numerical simulations and direct search for worst-case scenarios (e.g., Averkiev and Klevanny, 2010), the use of the probabilistic approach is a classic way to quantify the properties of extreme water levels and related

- risks. The relevant pool of literature contains substantial amount of work on both extreme water levels and their return periods (e.g., Purvis et al., 2008; Haigh et al., 2010; Arns et al., 2013) and on statistical parameters of water level variations (Serafin and Ruggiero, 2014; Fawcett and Walshaw, 2016). Similar probabilistic analysis has been extensively applied to average and extreme wave properties (e.g., Orimolade et al., 2016; Rueda et al., 2016), and properties of meteotsunamis
- (Geist et al., 2014, Bechle et al., 2015). In most occasions the drivers of coastal floodings are neither completely independent nor completely dependent on each other. This feature generates the necessity to consider multivariate distributions of their properties. Most often, the simultaneous occurrence of storm surges and large waves is considered (e.g., Hawkes et al., 2002; Wadey et al., 2015; Rueda et al., 2016b); occasionally including also an analysis of joint distributions of wave heights, periods and directions (Masina et al., 2015).
- Importantly, typical probability distributions of different contributors to extreme water levels may be fundamentally different. While the distribution of water levels is usually close to a Gaussian one (Bortot et al., 2000; Johansson et al., 2001; Mel and Lionello, 2014; Soomere et al., 2015), the probabilities of occurrence of different single wave heights are at best approximated either by a Rayleigh (Longuet-Higgins, 1952), Weibull (Forristall, 1978) or Tayfun distribution (Socquet-Juglard et al., 2005). The empirical probabilities of average or significant wave heights in various offshore conditions usually
- resemble either a Rayleigh or a Weibull distribution (Muraleedharan et al., 2007; Feng et al., 2014) while Pareto-type distributions are more suitable for the analysis of meteotsunami heights (Bechle et al., 2015). The total water level in semi-sheltered seas with extensive subtidal- or weekly-scale variability may contain two components, one of which has the classic quasi-Gaussian distribution whereas the other (storm surge) component has an exponential distribution and apparently mirrors a Poisson process (Soomere et al., 2015).
- Wave-driven local water level set-up is one of the classic phenomena at open ocean coasts. It may often provide as much as 1/3 of the total water level rise during a storm (Dean and Bender, 2006) and significantly contribute to extreme sea level events (Hoeke et al., 2013; Melet et al., 2016). Even though the physics of wave-set up (Longuet-Higgins and Stewart, 1964) is known for half a century, adequate parameterizations have been introduced (Stockdon et al., 2006) and many models take into account wave set-up to a certain extent (SWAN, 2007; Roland et al., 2009; Alari and Kõuts, 2012; Moghimi et al.,
- 2013), the contribution from this phenomenon apparently provides one of the largest uncertainties in modelling of storm surges and flooding (Dukhovskoy and Morey, 2011; Melet et al., 2013). This feature reflects the intrinsically complicated nature of the formation of this phenomenon. First of all, the set-up height strongly depends on the approach angle of waves at the actual breaker line. This angle is well-defined only if the coastline is almost straight, the nearshore is homogeneous in the alongshore direction and the wave field is monochromatic (Larson et al., 2010; Viška and Soomere, 2013; Lopez-Ruiz et

al., 2014; 2015). Generally, this angle is a complicated function of shoreline geometry, nearshore bathymetry, wave properties and possibly increased water level. Even if the basic wave properties (height, period and propagation direction) are perfectly forecast or hindcast at some nearshore location, evaluation of the further propagation of waves over nearshore bathymetry is a major challenge because, e.g., refraction properties change along with the changes to the local water level. Moreover, accurate wave hindcast and forecast are still a challenge in many water bodies and the outcome of modelling

- substantially depends on the used wind information (Nikolkina et al., 2014). Several studies have focused on maxima of wave set-up over extended areas (O'Grady et al., 2015; Soomere et al., 2013) or the maxima of the contribution from wave set-up to water level extremes (Pindsoo and Soomere, 2016). This problem is relatively simple on comparatively straight open ocean coasts that are fairly homogeneous in the alongshore direction and
- where the highest waves tend to approach the shore under relatively small angles. These features make it possible to use simplified schemes for evaluation of the impact of refraction and shoaling in the nearshore (e.g., Larson et al., 2010) or to assume that waves propagate directly onshore (O'Grady et al., 2015), and to reduce the problem to an evaluation of the properties of highest waves from a relatively narrow range of directions (Soomere et al., 2013). In areas with complicated geometry and especially in coastal segments where high waves may often approach under large angles to the shoreline it is
- necessary to take into account full refraction and shoaling in the nearshore (Viška and Soomere, 2013; Pindsoo and Soomere, 2015).

Even though high storm surges are often associated with severe seas, the development of high set-up depends on many details. It does not necessarily reach its maximum level in the entire coastal section affected by a storm and in most occasions the maximum storm surge and maximum wave set-up do not occur simultaneously (Pindsoo and Soomere, 2015).

Moreover, it seems characteristic for coastal areas with complicated geometry that each short segment has its own 'perfect storm' that creates the all-highest sum of storm surge and wave set-up (Soomere et al., 2013). These observations call for further analysis of the properties of the phenomenon of wave set-up. In this paper we focus on

certain features of statistical distributions of set-up heights along an about 90 km long coastal section with complicated geometry and containing segments open towards fairly different directions. The goal is to identify the typical shapes of distributions of simulated wave set-up heights and to analyse alongshore variability of these distributions. The test area is the

- distributions of simulated wave set-up heights and to analyse alongshore variability of these distributions. The test area is the vicinity of Tallinn Bay in the Gulf of Finland, the Baltic Sea. The paper is organised as follows. Section 2 introduces the method of evaluation of maximum set-up height for obliquely approaching waves and provides a short overview of the simplified wave model for rapid evaluation of wave time series, the forcing data for this model and the procedure of evaluation of properties of breaking waves. Section 3 presents an analysis of
- spatial variations of the extreme wave heights along the study area and estimates of the shape of empirical probability distribution of different set-up heights along the coast. This distribution substantially deviates from similar distributions of different wave conditions and exhibits unexpectedly large proportion of high set-up events. Several implications of the results are discussed in Section 4.

# 2 Methods and data

#### 2.1 Set-up height for obliquely incident waves

The classic concept of wave set-up (Longuet-Higgins and Stewart, 1962) relates the local increase in the water level with the onshore component of radiation stress. For small depth (incl. the area near the breaker line) the beaching waves can be described using the approximation of long waves and this component of radiation stress can be approximated as follows:

$$S_{xx} \approx \left(\frac{1}{2} + \cos^2\theta\right) E \quad , \tag{1}$$

Here  $S_{xx} \approx \cos \theta$  is the approach angle of waves with respect to onshore-directed normal to the shoreline,  $E = \rho g H_{rms}/8 = \rho g H_S/8$  is the wave energy,  $\rho$  is water density, g is acceleration due to gravity,  $H_{rms}$  is the root mean square wave height and  $H_S$  is the significant wave height. In ideal conditions of a plane impermeable beach the maximum

10 set-up height for waves propagating directly onshore ( $\theta = 0$ ) along a planar beach profile is (McDougal and Hudspeth, 1983)

$$\overline{\eta}_{\max} = \frac{5}{16} \gamma_b H_b \tag{2}$$

where  $\gamma_b = d_b / H_b$  is the breaking index.

15

5

If waves approach under a nonzero angle  $\theta$  with respect to the shore normal, the situation is much more complicated. Shi and Kirby (2008) argue that the water level set-down at the breaker line is invariant with respect to the approach angle. The theoretical expression for the deviation  $\eta$  of the water surface from the still level at a depth d in the surf zone of an impermeable beach with parallel straight bottom is (Hsu et al., 2006; Shi and Kirby, 2008; the power at  $\gamma_b$  in the first term at the right-hand side being corrected):

$$\frac{\eta}{H_b} = \frac{\gamma_b^2 \sin^2 \theta_b}{2\left(8 + 3\gamma_b^2 - 2\gamma_b^2 \sin^2 \theta_b\right)} \left[ \left(\frac{d}{H_b}\right)^2 - 1 \right] - \frac{3\gamma_b^2 - 2\gamma_b^2 \sin^2 \theta_b}{8 + 3\gamma_b^2 - 2\gamma_b^2 \sin^2 \theta_b} \left(\frac{d}{H_b} - 1\right) - \frac{\gamma_b^2}{16}.$$
(3)

The last term at the right-hand side of Eq. (3) represents water level set-down at the breaker line,  $H_b$  is the breaking wave 20 depth and  $\theta_b$  is the wave approach direction at breaking. The relative maximum set-up  $\overline{\eta}_{max}$ , counted from this water level, occurs at a 'depth'  $d = -\overline{\eta}_{max}$  :

$$\frac{\hat{\eta}_{\max}}{H_b} = \frac{\gamma_b^2 \sin^2 \theta_b}{2\left(\!8 + 3\gamma_b^2 - 2\gamma_b^2 \sin^2 \theta_b\right)} \left[ \left( \frac{-\hat{\eta}_{\max}}{H_b} \right)^2 - 1 \right] - \frac{3\gamma_b^2 - 2\gamma_b^2 \sin^2 \theta_b}{8 + 3\gamma_b^2 - 2\gamma_b^2 \sin^2 \theta_b} \left( \frac{-\hat{\eta}_{\max}}{H_b} - 1 \right).$$
(4)

For shore-normal waves the set-up with respect to the water level at the breaker line is  $\hat{\eta}_{\text{max}} = (3/8)\gamma_b H_b$  and the waterline is located at  $\overline{\eta}_{\text{max}}$  defined by Eq. (2). In the general case Eq. (4) is a quadratic equation with respect to  $q = \hat{\eta}_{\text{max}} / H_b$ :

$$q = \frac{\gamma_b^2 \sin^2 \theta_b}{2(8+3\gamma_b^2 - 2\gamma_b^2 \sin^2 \theta_b)} (q^2 - 1) + \frac{3\gamma_b^2 - 2\gamma_b^2 \sin^2 \theta_b}{8+3\gamma_b^2 - 2\gamma_b^2 \sin^2 \theta_b} (q+1).$$
(5)

This equation can be rewritten as

5 
$$\gamma_b^2 \sin^2 \theta_b q^2 - 16q + 6\gamma_b^2 - 5\gamma_b^2 \sin^2 \theta_b = 0$$
 (6)

and has two positive solutions for physically reasonable values  $0 < \gamma_b \le 2$ . As for very small incidence angles  $\theta_b \approx 0$  the physically relevant solution must be bounded (almost equal to  $q \approx 3\gamma_b^2/8$ ), the expression

$$q_{1} = \frac{16 - \sqrt{256 - 4\gamma_{b}^{2}\sin^{2}\theta_{b}\left(6\gamma_{b}^{2} - 5\gamma_{b}^{2}\sin^{2}\theta_{b}\right)}}{2\gamma_{b}^{2}\sin^{2}\theta_{b}}$$
(7)

provides the necessary solution. Eq. (7) deviates from a similar expression (30) of Hsu et al. (2006) by reasons discussed by 10 Shi and Kirby (2008). The maximum set-up height is thus

$$\overline{\eta}_{\max} = \left(q_1 - \frac{\gamma_b^2}{16}\right) H_b.$$
(8)

# 2.2 Wave time series in along the nearshore of the study area

We evaluate the shape and parameters of statistical distribution set-up heights along an about 80 km long coastal segment of Tallinn Bay and Muuga Bay (Fig. 1). The study area is an example of a wave-dominated micro-tidal region. The shoreline of

- this section of the northern shore of Estonia at the coast of the Gulf of Finland in the north-western Baltic Sea is locally almost straight (for scales up to a kilometre or two). Several relatively straight parts at Suurupi Peninsula and the area of Saviranna are open to the north. However, on larger scales (from a few kilometres) the coast contains large peninsulas and bays deeply cut into the mainland. The shores of these landforms are open to various directions. As the formation of wave set-up crucially depends on the wave height and the attack angle, this type of coastal landscape makes it possible to involve
- coastal sections with radically different magnitudes of set-up (see "climatology" of set-up heights in this area in Soomere et al., 2013).

The fetch length in the Gulf of Finland is >200 km for western and eastern winds but is generally below 100 km for all other wind directions. The all-time highest significant wave heights in the Gulf of Finland just a few tens of km to the north of the study area have exceeded 5.2 m (Tuomi et al., 2011). Strong north-north-western storms may generate significant wave

heights >4 m in the interior of Tallinn Bay (Soomere, 2005). The predominant strong wind directions in this region are

south-west and north-north-west. Eastern storms are less frequent but may generate as high waves as the western storms (Soomere et al., 2008). The varying mutual orientation of high winds and waves and the shoreline sections thus makes it possible to analyse potential variations in the distributions of set-up heights.

We employ time series of wave properties (significant wave height, wave period and propagation direction) reconstructed using the wave model WAM and one-point high-quality wind information from the vicinity of the study area for 1981–2016.

- The wave model is set-up in a triple nested version with the resolution of the innermost grid about 470 m (Soomere, 2005). Following the experience of using this model in the Baltic Sea and Finnish archipelago conditions (where it is important to adequately represent the wave growth in low wind and short fetch conditions) (Tuomi et al., 2011; 2012), the model uses an increased frequency range of waves up to 2.08 Hz. The ignoring of the presence of sea ice may lead to a certain
- overestimation of the overall wave energy in the region but apparently does not significantly distort the shape of distributions of set-up heights and the variation of these distributions along the shoreline. The model is implemented using a simplified scheme for rapid reconstructions of long-term wave statistics. Wave computations are speeded up by replacing long-term calculations of the sea state by an analysis precomputed maps of wave properties. This simplification relies on a favourable feature of the study area. Namely, wave fields rapidly become saturated
- and have relatively short memory in the study area (Soomere, 2005). Consequently, a reasonable reproduction of wave statistics is possible by assumption that an instant wave field in Tallinn Bay is a function of a short section of the wind dynamics. This assumption justifies the splitting of the calculations into a number of short independent sections with a duration of 3–12 hours. As details of the particular model set-up, the used bathymetry, the implementation, and validations of the model outcome have been repeatedly discussed, the reader is referred to (Soomere, 2005; Soomere et al., 2013) for
- further information.

The described approach, however, makes it possible to circumvent one of the major issues of replication of the Baltic Sea wave fields, namely, the problems with quality and frequent inconsistency of modelled wind data sets (Nikolkina et al., 2014). The quality of wave hindcast primarily depends on the adequacy of the wind information. In particular, wave set-up is intrinsically very sensitive with respect to the wave propagation direction. It is therefore crucial to force the wave model with

- correct information about wind directions. This is an issue in the Gulf of Finland because atmospheric models often fail to reproduce wind directions in this water body (Keevallik and Soomere, 2010). To overcome this issue, we use wind data from an offshore location in the central part of this gulf. The wind recordings at Kalbådagrund (59°59' N, 25°36' E, a caisson lighthouse located on the top of a shoal far offshore) are known to impeccably represent marine wind properties. Even though this site is located at a distance of some 60 km from the study area, it is expected to correctly record wind properties
- in the offshore that are mostly responsible for the generation of surface waves.
  - The entire simulation interval 1981–2016 contained 103 498 wind measurement instants with a time step of 3 h. As this resolution of wind measurements was employed for more than two decades, we selected analogous data also from the newer higher-resolution recordings. In about 9000 cases (less than 10%) either wind speed or direction was missing. These time

instants were excluded from the further analysis. As some of these instants involved quite strong winds, the analysis may underestimate the highest wave set-up events.

#### 2.3 Nearshore refraction and shoaling

- The nearshore grid cells selected for the analysis (Fig. 1) are located in at least 4 m deep area in order to avoid massive wave 5 breaking already at the formal calculation location. Some of the cells are located in even deeper locations with water depth 20–27 m. The presence of various underwater features and inhomogeneities in the study area means that shoaling and refraction may considerably impact the wave fields even along the relatively short routes (normally about 1 km) from the model grid cell until the breaking line. As the predominant storm directions are the south-east, north-north-west and east, in most occasions high waves approach some of the selected grid cells under large angles with respect to the shore normal.
- 10 Therefore, it is not acceptable to assume that the incidence angles are small and simplified approaches to replicate the changes in the immediate nearshore (Lopez-Ruiz et al., 2014; 2015) and even advanced approximations (Hansen and Larson, 2010) may fail. For this reason we calculate the joint impact of shoaling and refraction of approaching waves in the framework of the linear wave theory (Viška and Soomere, 2013; Soomere et al., 2013).
- As usual, it is assumed that the numerically evaluated wave field for each time instant is monochromatic and characterised by the numerically simulated significant wave height  $H_0$ , peak period and mean approach direction  $\theta_b$  (with respect to the onshore-directed normal to the shoreline) that are evaluated at the centre of each selected grid cell. Similarly, it is assumed that the nearshore seabed is plane with isobaths strictly parallel to the shoreline and that breaking waves are long waves. Then the wave height  $H_b$  at the breaking line can be found as the smaller real solution of the following algebraic equation of 6th order (Viška and Soomere, 2013; Soomere et al., 2013):

20 
$$\frac{H_b^5 g}{H_0^4 \gamma_b} \left( 1 - \frac{g H_b}{\gamma_b} \frac{\sin^2 \theta_0}{c_{f0}^2} \right) = c_{g0}^2 \left( 1 - \sin^2 \theta_0 \right).$$
(8)

Here  $c_g$  is the group speed,  $c_f$  is the phase speed and the subscripts "0" and "b" indicate the relevant value at the particular wave model grid cell and at the breaker line, respectively. The set of assumptions is completed with the common notion that the breaking index is  $\gamma_b = H_b/d_b = 0.8$ .

Several earlier studies of extreme wave set-up heights (Soomere et al., 2013; Pindsoo and Soomere, 2015) have taken into account only waves that approach the coast under angles  $\pm 15^{\circ}$  with respect to the shore normal. This assumption is evidently valid at the open ocean coasts where waves usually approach the shore under relatively small angles. However, in specific conditions of semi-sheltered basins with short fetch and with several strong wind directions, this assumption may become unusable. To evaluate its applicability and to estimate distortions to the distributions of different set-up heights, we repeated calculations based on this selection of wave fields.

# **3 Results**

#### 3.1 Maximum set-up heights

The phenomenon of wave set-up is only meaningful if large waves are propagation from the open sea towards the shore. Differently from open ocean coasts where swells always create certain set-up, in sheltered sea areas with complicated

- geometry swells may be infrequent and wind waves often propagate from the nearshore towards the open sea. Wind regime of the study area consists of superposition of four relatively frequent wind systems (Soomere et al., 2008). The most frequent wind direction here is from the south-west (that is, from the mainland to the sea). Still, the proportion of wave systems with the average propagation to the onshore is about 60% (Fig. 2). The only exception is grid cell 107 (Fig. 1) between Viimsi Peninsula and the island of Aegna that is sheltered for almost all directions. The statistical properties of wave set-up
- phenomena discussed below thus represent 40,000–70,000 examples of wave fields for all other cells. We start from a comparison of maximum set-up heights evaluated using the above-described approach and a simpler method (Soomere et al., 2013) that took into account wave fields that propagated almost directly onshore (±15° with respect to the shore normal). The two methods lead to comparable results for all coastal segments that are relatively open to the Gulf of Finland (Fig. 3). As expected, the simplified method gives systematically higher results for coastal segments that are
- relatively open to the Gulf of Finland (Fig. 3). The differences in the maximum set-up heights are mostly close to 0.2–0.3 m, but in some occasion just a few cm. The largest maxima are relatively different. The two methods give different estimates also for the sheltered areas and particularly for 'side' sections of deeply cut bays (that is, sections to which waves approach under a relatively large angle). Interestingly, estimates using the method that fully calculates the radiation stress are remarkably higher in these sections than those with the simplified method (that tends to
- give higher values in all other areas). This difference indicates that the impact of refraction is substantial. It is thus likely that refraction may redirect waves so that beaches that even seemingly well sheltered geometrically may at times receive remarkable amounts of wave energy. In other words, the impact of refraction often overrides here the effect of geometric blocking of waves by changing orientation of the coastline (cf Caliskan and Valle-Levinson, 2008).
- The maximum differences in set-up heights for such (seemingly geometrically sheltered) sections are often 0.2–0.3 m and reach up to 0.6 cm in a few occasions. Such a strong impact of refraction is not usual but also not unique in coastal areas. It is usually thought to be responsible for a local increase in wave heights not only in the Baltic Sea (Soomere, 2003) but also in extreme ocean conditions (Babanin et al., 2011). The major differences in the two sets of estimates of set-up heights clearly signal that the use of simplified methods and taking into account only waves that propagate almost onshore may lead to substantial underestimation of various wave-driven hazards.
- Notice that some differences of the latter set of results from those presented in (Soomere et al., 2013; Pindsoo and Soomere, 2015) stem from the different time periods used in the calculations. Simulations for 1981–2010 indicate that the maximum set-up heights in coastal areas open to the east were mostly created in the 1980s (Soomere et al., 2013) even though the

maximum wave heights occurred starting from the mid-1990s. This feature may be related to a change in the strong wind directions so that eastern storms became weaker.

However, there is increasing evidence that this process has reversed and strong eastern storms have returned to the area. The first evidence of this change is the that the all-time highest significant wave height 5.2 m was recorded in the Gulf of Finland

for the second time during an extreme eastern storm on 29–30 November 2012 (Pettersson et al., 2013). Pindsoo and Soomere (2015) observed that many new all-time highest set-up events apparently occurred in locations open to the east. This process evidently continues and has led to generation of all-highest simulated waves in a number of locations at the eastern Viimsi Peninsula near Leppneeme (Fig. 4).

# 3.2 Frequency of occurrence of set-up heights

- Somewhat surprisingly, the empirical distributions of occurrence of set-up heights are clearly non-Gaussian in all locations of the study area (Fig. 5). Even though the replication of wave propagation directions by the wave model and the impact of refraction may suffer from insufficient resolution of both the wind information and wave model, the presented distributions exhibit a similar shape for the entire study area. Their appearance clearly differs from all usual distributions of the magnitude of wave phenomena such as the classic (Rayleigh) distribution of single wave heights (Longuet-Higgins, 1952), Tayfun
- distribution of the heights of largest waves, the Weibull family of distributions for the occurrence of various wave conditions, or the Rayleigh distribution for run-up of (narrow-banded) Gaussian wave fields (Didenkulova et al., 2008). Therefore, none of these distributions can be used for the approximation of wave set-up heights.
  Also, as the emperators of distributions of set up heights in log linear coordinates is clearly concerned along the

Also, as the appearance of distributions of set-up heights in log-linear coordinates is clearly concave upwards along the entire study area, even the exponential distribution (that describes, e.g., storm surges in the study area, Soomere et al., 2015)
is not suitable for their description. This appearance suggests that the background process is not a Poisson one (that would

- lead to basically linear shape of the distribution in question in log-linear coordinates). Importantly, this concave-up appearance of the distributions of set-up heights means that very large set-up events are systematically much higher and/or occur much more frequently than similar events of highest storm surges.
- To further explore the shape of the distributions of set-up height and its possible variations along the shoreline we assume that these distributions belong to the family of general exponential distributions. The overall appearance of these empirical distributions in log-linear coordinates suggests that their shape can be, as a first approximation, matched with a quadratic polynomial  $ap^2 + bp + c$ . The values of the coefficient *a* at the leading term are exclusively negative and predominantly close to -1 in the entire study area (Fig. 6). The absolute values of this parameter are clearly larger than 1 only along the eastern coast of several bays deeply cut into mainland (Tallinn Bay, Jõesuu Bay) and in one location of the interior of Kopli
- 30 Bay (Fig. 1). This feature indicates that the relevant distribution is close to the classic inverse Gaussian (Wald) distribution. and its basic shape is (Folks and Chikara, 1978)

$$P \sim \exp\left(-p^2\right). \tag{9}$$

Interestingly, the coefficient a at the leading term of the approximating polynomial varies insignificantly along the study area. More importantly, its variations are uncorrelated with the values of maximum set-up heights along the study area. This feature signals that the basic features of the distribution in question are invariant with respect to the properties of local wave climate, incl. the shape of the relevant Weibull distribution for different wave conditions (that largely varies along the study area of other study area are shown in Source 2005). This conjusture is supported by comparatively small varies of the values of other study area.

area as shown in Soomere, 2005). This conjecture is supported by comparatively small variations of the values of other parameters in the polynomial approximation (Fig. 6b,c). The values of b are all positive and mostly in the range of 2–4. The values of parameter c are typically positive and between 0 and 3; however, a few negative and even larger positive values occur in single cells.

#### **4** Discussion

- The performed analysis first of all reveals that numerical estimates of maxima of wave set-up heights are relatively sensitive with respect to the particular way of evaluation of the impact of radiation stress and the transformation of wave properties in the nearshore. The magnitude of the related effects substantially depends on the appearance of bathymetry. The impact of refraction can easily override the purely geometric effects of shoreline orientation changes and redirect substantial levels of wave energy into seemingly sheltered shore sections. This feature calls for the necessity of using high-resolution information
- about wind (incl. wind directions) and bathymetry together with advance methods for the evaluation of propagation and impact of radiation stress in the nearshore in operational and hindcast models of coastal flooding. The core message of the analysis is that the basic shape of probability distribution function of different set-up heights is concave upwards in log-linear plot. The appearance of this distribution varies insignificantly along the study area. As this area contains a variety of sections open to different directions and with radically different wave properties, the qualitative
- shape in question is basically invariant with respect to the properties of local wave climate. Further, the empirical distributions of wave set-up heights can be adequately approximated with a family of exponential distributions  $P \sim \exp(-p^2)$  with a negative coefficient at the quadratic approximation of the exponent. The relevant distributions are known as inverse Gaussian (Wald) distributions. Interestingly, the coefficient at the leading term of such a quadratic approximation is close to -1 and varies insignificantly along the entire study area (Fig. 6a). This feature once more
- confirms that the established distribution is universal for wave set-up heights even though its details may obviously depend on local conditions. As the coefficient at the linear term of this quadratic approximation is clearly nonzero (Fig. 6b), the distribution in question does not reduce to a Lévy distribution.

This result is intriguing because sensible approximations of inverse Gaussian (Wald) distributions are scarce in descriptions of geophysical phenomena. Perhaps the most well-known example of the use of a Wald distribution is to describe the time a

30 Brownian motion (with positive drift) takes to reach a fixed positive level. Other examples include statistical properties of soil phosphorus (Manunta et al., 2002), long-distance seed dispersal by wind (Katul et al., 2005) or some models of failure (Park and Padgett, 2005).