# Peer review of "Inverse Gaussian distribution of wave set-up heights along a shoreline with complicated geometry"

_Earth System Dynamics, 2016_

## Referee Comment (RC1) · Anonymous Referee #1 · 10 Feb 2017

Review of

Inverse Gaussian distribution of wave set-up heights along a shoreline with complicated geometry

by Tarmo Soonere and Katri Pindso

**Synopsis**

The paper investigates the probability density distribution (PDF), and especially its largest values, of wave run-up heights. Run-up is derived from wave heights, which

are obtained from a wave model (WAM). The wave model is driven by winds from just one measurement location, which is considered representative for the Gulf of Finland. Run-up heights are evaluated at 175 grid points with a distance of 470 m along the Gulf of Finland coast of Estonia. The chosen coastline is very complicated, with stretches of it at any angle with respect to the predominant wave and wind directions. The run-up heights are found to be distributed according to a an inverse Gaussian distribution, implying that the highest run-ups can be much higher than the highest one ever observed. Due to refraction, very high run-ups can also be found at seemingly sheltered locations.

**Discussion**

Although the topic and the results presented in this paper are relevant and interesting, I cannot recommend publication of the paper in its present form. There are two reasons, namely

- The presentation is unclear. Several times I had to re-read parts of the text, and I am still not sure whether I understood what was written. The instances in question are detailed below. A thorough re-writing of the paper is necessary.

- More importantly the analysis method used is inadequate. A fit of the whole distribution is sought in order to infer the behaviour of its tail. This approach is notoriously error-prone because the precise shape of the fitted distribution depends on the few cases in its tail. To infer the behaviour of the extreme values, extreme value statistics should be applied, e.g., by fitting a GEV (or, if possible, a Gumbel distribution) to the annual maxima. Other possibilities are POT or r-largest. The book of Coles (An Introduction to Statistical Modeling of Extreme Values, Springer, 2001) discusses them. The paper of Van den Brink and Können (Int. J. Climatol., 2009, DOI: 10.1002/joc.2047) discusses methods to test
the adequacy of the chosen fit if a lot of data points are available. This is here the case (175 points along the coast), and according to the claim of the authors they all follow the same PDF. This should be reflected by the test.

**Detailed comments**

**p 1, l 16** that → than

**p1, l 28** insert a comma after *drivers*

**p 4, l 3** with → to

**p 4, eq. (1)** The symbol $S_{xx}$ is not used in the following. That might be ok, but I was wondering why is it then introduced?

**p 4, l 7** I do not understand why $S_{xx}$ should approximately equal $\sin\theta$, and why it is the approach angle. The angle is $\theta$, isn't it?

**p 4, l 17** at → of

**p 4, eq. (4)** I do not understand the difference between $\overline{\eta}$ and $\hat{\eta}$, nor do I understand the sentence preceding the equation. Probably a sketch of the situation would help.

**p 6, l 13** insert *of* before *precomputed*

**p 6, l 12-20** I am not sure whether I get this correctly. What I understand is that you run WAM for a lot of wind cases. Then you run through your 100,000 measurements, determine which of your pre-run cases corresponds to it, and take the result of that calculation for the current situation. Is this really what you are doing?

**p 8, l 3** propagation → propagating

**p 8, l 13-15** I do not understand. How can a sub-set of cases (small incident angel) give higher values than the full set? Why did you expect it? Furthermore, the second and third sentence of this para directly contradict each other. The former says results are equal, the latter the opposite.

**p 8, l 16** What do you mean by *relatively different*?

**p 8, l 21** change to "... so that even beaches that are seemingly well sheltered ..."

**p 9, l 3-8** This para is on a different topic (decadal variability) and interrupts the argument to be made. Delete or move to an extra (sub)section.

**p 9, l 10** on → at

**p 9, l 18** *concave upward* - is this significant? I mean, are there enough points out in the tail of the distribution? $10^{-2}$% of 100,000 is 10, and $10^{-3}$% is just 1, so your conclusion hinges on very few points. Are you sure that it isn't just sampling? - See my main concern #2 above.

**p 9, l 27** What is $p$? $\ln \eta$?

**p 10, 1st para** Can you give error bounds for coefficients a, b, and c?

**p 10, l 11** Which particular way?

**p 10, l 15** advance → advanced

**p 10, l 17** insert *the* before *probability*

**p 10, l 28-32** Why is it intriguing that the inverse Gaussian is rare in geophysics? Is it more predominant in other areas?

**p 16, 2nd line of legend** about → of about; and *at* → *of*

[Figure]

**p 17, 2nd line of legend fig. 3** *these* → *those*; and *rhombi*: I think that this shape is usually denoted as *diamond*.

**p 19, 2nd line of legend fig. 5** *the first gap* - what is this? It is nowhere referred to and explained in the main text.

---

## Referee Comment (RC2) · Anonymous Referee #2 · 4 Mar 2017

The paper addresses a particularly interesting point and that is why the nature and frequency of high water level (surge and setup) events is different to and often more extreme than expected. I found the paper confusing in places, and I encourage the authors to rewrite the paper with clarity of methodology in mind.

I am not an expert in extreme value statistics, and leave it to other referees and contributors to comment on that aspect of the paper. The results, however, point to a number of important considerations, in particular the sensitivity to the application of radiation stress and the data requirements for its appropriate consideration. The next (but difficult) step would be some field measurements to compare to the model results.

There are numerous grammatical errors that need to be corrected. I would be happy to

undertake an edit, but they are too numerous to do without having access to an easily editable version of the paper (not pdf).

I ask that the authors review their axis labels and figure captions. eg

Figure 2 - should the y axis be 'onshore' and it is not a % Figure 3 - units on y axis

I think the paper is worthy of publication subject to rewriting, perhaps more than minor, but not major.

———————————————————

---

## Author Comment (AC1) · 24 Mar 2017

We very much appreciate the overall positive attitude of referee to our manuscript and thank him or her for particularly useful comments. Please find below the detailed reply to each point raised by the referee. The comments, questions and suggestions of the referee are presented in italics.

Referee 1

**Discussion – General comments**

- *The presentation is unclear. /—/ A thorough re-writing of the paper is necessary.*

[Figure]

We apologise for this shortage and agree that many sections of the manuscript are too compressed and partially use jargon of engineering. We are thankful for the detailed comments and shall be happy to remove all these deficiencies in the revised version. We only would like to indicate a possible partial source of this opinion of Referee 1. Namely, the review seems to reflect the analysis of wave run-up whereas we address the process of wave set-up. These phenomena are conceptually different. Wave run-up is a rhythmic motion of waterline when water carried by single waves rushes up the coast for a short time. Wave set-up creates an average elevated water level, which occurs due to the release of momentum carried by waves in the process of wave breaking.

- *More importantly the analysis method used is inadequate. A fit of the whole distribution is sought in order to infer the behaviour of its tail. This approach is notoriously error-prone because the precise shape of the fitted distribution depends on the few cases in its tail. To infer the behaviour of the extreme values, extreme value statistics should be applied, e.g., by fitting a GEV (or, if possible, a Gumbel distribution) to the annual maxima. Other possibilities are POT or r-largest. The book of Coles (An Introduction to Statistical Modeling of Extreme Values, Springer, 2001) discusses them.*

We believe that this comment reflects a simple misunderstanding. We do not consider the behaviour of the tail of the distribution. This is stated on lines 25–26 of p. 3 of the manuscript: "The goal is to identify the typical shapes of distributions of simulated wave set-up heights and to analyse alongshore variability of these distributions." We are of the opinion that this question is worth of research because the distribution in question seems to qualitatively differ from the commonly used distributions in the field (Gaussian and Poisson distributions). We do not discuss how the extreme values of wave set-up heights are distributed. Also, we make no quantitative attempt to evaluate the possible extremes based on the resulting shape of the distributions. We are well aware that

extreme values of various phenomena generally follow a Weibull, Frechet or Gumbel distribution, or, more generally, a Generalized Extreme Value (GEV) distribution as described in the nice book of Coles (2001). Instead, we discuss the general shape of the distribution of set-up heights, more specifically, make an attempt to identify into which class of distributions it belongs. This question is irrelevant from the viewpoint of evaluation of the distribution of set-up maxima but may become crucial in studies of joint extremes of processes that drive coastal floodings. It seems that, for example, in the Baltic Sea the entire water volume of the sea follows a Gaussian distribution and storm surge is a Poisson process (Soomere et al., 2015. Separation of the Baltic Sea water level into daily and multi-weekly components. Continental Shelf Research, 103, 23–32). The observation that the third core component of coastal flooding, wave set-up, reflects an inverse Gaussian process, may severely complicate the analysis of distributions and extremes of their joint impact.

- *The paper of Van den Brink and Können (Int. J. Climatol., 2009, DOI: 10.1002/joc.2047) discusses methods to test the adequacy of the chosen fit if a lot of data points are available. This is here the case (175 points along the coast), and according to the claim of the authors they all follow the same PDF. This should be reflected by the test.*

We believe that this remark is a little bit out of the scope of our study. While we are interested in the overall shape of the distribution (that is not known beforehand) and do not pay any attention to extremes or outliers, Van den Brink and Können (2011) start from an assumption that the block maxima follow a Gumbel distribution and then check whether the presence of outliers may ruin the applicability of this extreme value distribution. Also, in some examples Van den Brink and Können (2011) assume that some parameters of a Generalized Extreme Value (GEV) are constant over Europe. We do not set any constraints for the parameters of the function that is used to approximate the appearance of the exponent in the selected (quite wide) class of distributions.
However, we agree with the Referee that a certain kind of test is necessary to quantify the credibility of our results. Thus, we shall provide confidence intervals of the leading coefficient of the quadratic polynomial in the revised manuscript to show that virtually in all cases this coefficient is negative with a high confidence. We also consider an option to include a classic test of the adequacy of the chosen fit.

**Discussion – General comments**

We are grateful for these comments and shall of course take all these into account in the revised version. Our apologies for a jam that renders line 7 on p. 4 meaningless. Somehow we deleted a couple of lines that explain the role of radiation stress in the process and how one can derive an estimate of set-up height from this quantity.

Also, we have been careless while using the symbol $\eta$ for set-up height with different embellishments on lines 19–20 (p. 4) and Eq. (4). This will definitely be corrected and the explanation of math will be expanded. It is also a good idea to insert a sketch of the entire situation and used variables.

- *p 6, l 12-20 I am not sure whether I get this correctly. What I understand is that you run WAM for a lot of wind cases. Then you run through your 100,000 measurements, determine which of your pre-run cases corresponds to it, and take the result of that calculation for the current situation. Is this really what you are doing?*

Yes, basically this is exactly what we are doing. The actual scheme contains the response to many specific situations with the goal to adequately reconstruct wave properties for relatively high wind speeds.

- *p 8, l 13-15 /—/ How can a sub-set of cases (small incident angle) give higher values than the full set? Why did you expect it?*

Wave set-up height is very sensitive with respect to the approach angle of waves and decreases rapidly for waves that approach the shore under larger angles. Therefore, the highest set-up events are normally created by waves that propagate (almost) directly onshore. This is why in earlier studies we only considered waves with propagation direction $\pm 15$ degrees from the shore normal. In earlier studies we did not resolve the impact of changes in the wave propagation direction within the surf zone on the set-up height. As a result, the height of set-up of waves approaching under angles of 10–15 degrees with respect to shore normal was often overestimated. In this manuscript we use an improved scheme to evaluate set-up height according to (Hsu et al., 2006; Shi and Kirby, 2008). We shall explain these issues in more detail in the revised version.

- *Furthermore, the second and third sentence of this paragraph directly contradict each other. The former says results are equal, the latter the opposite. p 8, l 16 What do you mean by relatively different?*

We actually say that they are "comparable". However, we understand that our original wording is misleading and we shall rephrase the sentences.

- *p 9, l 3-8 This paragraph is on a different topic (decadal variability) and interrupts the argument to be made. Delete or move to an extra (sub)section.*

Thank you for this observation; we shall rearrange the text in the revised version.

- *p 9, l 18 concave upward – is this significant? I mean, are there enough points out in the tail of the distribution? $10^{-2}$% of 100,000 is 10, and $10^{-3}$% is just 1, so your conclusion hinges on very few points. Are you sure that it isn't just sampling? See my main concern #2 above.*

The concave appearance of the geometric representations of all distributions is an unexpected feature. It is well known that water levels usually follow a distribution similar to a Gaussian one and wave run-up normally follows either a Gaussian or a Weibull one. In other words, all distributions of the impact phenomena that govern water level in the nearshore and the location of the waterline follow distributions with a positive or zero coefficient a at the leading term of the quadratic approximation of the relevant exponent. The geometric representations of such distributions are either concave downward (Gaussian, Weibull) or appear as straight line (Poisson/exponential) in semilogarithmic coordinates. Now we have a different shape, exclusively concave upward for all segments of the coast with extremely complicated geometry. Technically, the sign of the coefficient a represents the sign of the curvature of the relevant line. The values of this coefficient are negative for all segments of the coast. To make sure the potential impact of sampling, we recalculated the coefficients a, b and c for the main body of the distribution in the range of 0.01–10%. The results are basically the same. We mention once more that we do not draw any quantitative conclusions about the behaviour of extreme set-up heights based on our analysis. We only stress that it is likely that very high set-up events may occur more frequently than it could be derived from a Gaussian statistics.

- *p 10, 1st para Can you give error bounds for coefficients a, b, and c?*

We shall be happy to present also error bounds for these coefficients

- *p 9, l 27 What is p?*

It is just dummy variable, used to express the type of approximation.

- *p 10, l 28-32 Why is it intriguing that the inverse Gaussian is rare in geophysics? Is it more predominant in other areas?*

Actually, we tell that our result is intriguing because the inverse Gaussian is rare in geophysics. To our knowledge, inverse Gaussian (Wald) is scarcely used in other fields of science and engineering.

- *p 19, 2nd line of legend fig. 5 the first gap - what is this? It is nowhere referred to and explained in the main text.*

We apologise for not explaining the technique in sufficient detail. To avoid problems with sampling, we only approximated the distribution from its largest values (very small set-up) up to the first value, for which the data set had zero occurrence. We did so even if some even higher values of set-up height occurred many (tens of) times.

---

## Author Comment (AC2) · 24 Mar 2017

We very much appreciate the overall positive attitude of referee to our manuscript and thank him or her for particularly useful comments. Please find below the detailed reply to each point raised by the referee. The comments, questions and suggestions of the referee are presented in italics.

Referee 2

- */—/ I found the paper confusing in places, and I encourage the authors to rewrite the paper with clarity of methodology in mind.*

[Figure]

Referee #1 also made this point. We agree that the material of the paper is too compressed in many sections and additional information and explanations will make it more clear and transparent.

- */—/ The next (but difficult) step would be some field measurements to compare to the model results.*

This is also our hope – to arrange measurements of wave set-up in some hot spots of the study area. Today we have to rely on anecdotal evidence.

- */—/ There are numerous grammatical errors that need to be corrected. /—/ they are too numerous /—/ I ask that the authors review their axis labels and figure captions. E.g. Figure 2 - should the y axis be 'onshore' and it is not a /*

Thank you for indicating these issues, we shall correct them in the revised version.